

# Variability and long-term changes of tropical cold point temperature and water vapor

Mona Zolghadrshojaee[1], Susann Tegtmeier[1], Sean M. Davis[2], Robin Pilch Kedzierski[3]

[1]Institute of Space and Atmospheric Studies, University of Saskatchewan, Saskatoon, Canada
[2]National Oceanic and Atmospheric Administration Chemical Sciences Laboratory, Boulder, CO 80305, USA
[3]Meteorological Institute, Universität Hamburg, Hamburg, Germany

*Correspondence to*: Susann Tegtmeier (susann.tegtmeier@usask.ca)

**Abstract.** The tropical tropopause layer (TTL) is the main gateway for air transiting from the troposphere to the stratosphere and therefore impacts the chemical composition of the stratosphere. In particular, the cold point tropopause, where air parcels

encounter their final dehydration, effectively controls the water vapor content of the lower stratosphere. Given the important role of stratospheric water vapor for the global energy budget, it is crucial to understand the long-term changes in cold point temperature and their impact on water vapor trends.

Our study uses GNSS-RO data to show that, there has been no overall cooling trend of the TTL over the past two decades, in contrast to observations prior to 2000. Instead, the cold point is warming, with the strongest trends of up to 0.7 K/decade during

boreal winter and spring. The cold point warming shows longitudinal asymmetries with the smallest warming over the central Pacific and the largest warming over the Atlantic. These asymmetries are anti-correlated with patterns of tropospheric warming, and regions of strongest cold point warming are found to show slight cooling trends in the upper troposphere. Overall, the here identified warming of the cold point is consistent with model prediction under global climate change, which attributes the warming trends to radiative effects. The seasonal signals and zonal asymmetries of the cold point temperature and height

trends, on the other hand, seemed to be related to dynamical responses to enhanced upper tropospheric heating, changing convection, or trends of the stratospheric circulation.

Water vapor observations in the TTL show mostly positive trends for 2004-2021 consistent with cold point warming. We find a decrease of the amplitude of the cold point temperature seasonal cycle by ~7% driving a reduction of the seasonal cycle in 100 hPa water vapor by 5-6%. Our analysis shows that this reduction in the seasonal cycle is transported upwards together

with the seasonal anomalies and has reduced the amplitude of the well-known tape recorder over the last two decades.

## 1 Introduction

The tropical tropopause layer (TTL) is the transition region between the turbulent, moist troposphere and the stable, dry stratosphere (Fueglistaler et al., 2009). The TTL acts as the main gateway for air entering the stratosphere and extends from the region of strong convective outflow near 12–14 km to the highest altitudes reached by convective overshooting events,

around 18 km (e.g., Highwood and Hoskins, 1998; Folkins et al., 1999; Fueglistaler et al., 2009). In consequence, air masses



in the TTL are controlled by numerous processes on a wide range of lengths and timescales and show dynamical and chemical properties of both the troposphere and the stratosphere. Monitoring long-term changes in the TTL is crucial as the interactions among circulation, convection, trace gases, clouds, and radiation make this region a key player in radiative forcing and chemistry–climate coupling (e.g., Mote et al., 1996; Holton and Gettelman, 2001; Fueglistaler et al., 2011; Randel and Jensen,

2013). Atmospheric radiative fluxes in the TTL are particularly sensitive to temperatures, water vapor concentrations as well as the presence of thin cirrus clouds and convective anvils (Wang et al., 1996; Sassen et al., 2009). Theoretical arguments suggest that the TTL radiative balances can in turn impact altitudes of deep convective outflow (Hartmann and Larson, 2002; Harrop and Hartmann, 2012).

An important characteristic of the TTL is the change of the negative vertical temperature gradient of the troposphere into the

positive temperature gradient of the stratosphere. The vertical location of the temperature minimum is termed the Cold Point Tropopause (CPT; Highwood and Hoskins, 1998). Air parcels transitioning from the troposphere to the stratosphere encounter final dehydration at these minimum temperatures and in consequence, the cold point tropopause effectively controls the water vapor content of the lower stratosphere (Randel et al., 2004; Fueglistaler et al., 2009; Randel and Park, 2019). Stratospheric water vapor plays an important role in the global energy budget with increasing stratospheric water vapor cooling the

stratosphere and heating the troposphere (De F. Forster and Shine, 1999; Smith et al., 2001) and can impact stratospheric ozone loss (Vogel et al., 2011).

The temperature structure of the TTL is determined by the interplay of tropospheric and stratospheric processes. Ongoing strong convection across continents and the western Pacific Ocean contributes significantly to the convective equilibrium that governs the temperatures in the lower TTL. In particular, the connection between El Niño Southern Oscillation (ENSO) and

TTL temperatures via convective activities can cause large interannual variations in the latter (Yulaeva and Wallace, 1994). The enhancement of deep convection over the tropical West Pacific is caused by ENSO's negative phase (i.e., La Niña) which is accompanied by TTL cold anomalies right above (Gettelman et al., 2001). Similar TTL cold anomalies occur over the central Pacific during the positive phase (El Niño).

Conversely, the Brewer-Dobson circulation (BDC) with large-scale tropical upwelling, which keeps temperatures well below

radiative equilibrium, affects lower stratospheric temperatures. The BDC has a distinct annual cycle due to the seasonality of extratropical planetary wave driving, which imprints a pronounced seasonal cycle in upper TTL temperature. Significant interannual variations in upper TTL temperature are driven by the Quasi-Biennial Oscillation (QBO; Baldwin et al., 2001), a descending pattern of alternating zonal winds. The QBO temperature variations oscillate between the easterly and westerly shear phases following thermal wind balance. Easterly wind shear results in increased upwelling and cold anomalies, while

westerly wind shear results in decreased upwelling and warm anomalies (Plumb and Bell, 1982; Baldwin et al., 2001). If the QBO easterly phase occurs during boreal winter, seasonal mean deep convection over the western Pacific will be strengthened (Collimore et al., 2003; Liess and Geller, 2012) further amplifying the QBO cold anomalies in this region (Tegtmeier et al., 2020b). When compared to the equivalent stratospheric changes (±4 K at 25 km), the QBO temperature signal in the TTL is relatively small (±0.5 K at the CPT) and peaks between 10°S and 10°N (Randel and Wu, 2015).



Volcanic eruptions have long been recognized as major natural events capable of influencing global climate including TTL temperature. For the three major volcanic eruptions that have occurred since the late 1950s, analysis of observational and reanalysis data has revealed warming signals in the tropical lower stratosphere (Free and Lanzante, 2009; Fujiwara et al., 2015) and upper TTL (e.g., Krüger et al., 2008; Fujiwara et al., 2015; Tegtmeier et al., 2020a)

Given the important role that the TTL plays in the climate system and the chemical composition of the stratosphere, it is crucial

to understand long-term changes in TTL temperature. In particular, trends of the cold point temperature are of interest since they can lead to changes in stratospheric water vapor. Long-term changes of the TTL temperatures are regulated by complex coupling of chemical, dynamic, and radiative processes (Fueglistaler et al., 2009; Grise and Thompson, 2013). Changes in Greenhouse Gas (GHG) concentrations (e.g., Xie et al., 2008), the strength of the BDC (e.g., Wang et al., 2013; Fu et al., 2015), convection (e.g., Nishimoto and Shiotani, 2012), and Sea Surface Temperatures (SSTs; e.g., Hu et al., 2014) all have

an impact on TTL temperatures. Among these factors, the SST and GHG changes as well as convective activities play a key role in the long-term cold point temperature changes (e.g., Rosenlof and Reid, 2008; Xie et al., 2014).

Different data sources exist for deriving trends of TTL temperature. An extended satellite temperature record based on the Stratospheric Sounding Unit (SSU), the Advanced Microwave Sounding Unit (AMSU-A), the Microwave Limb Sounder (MLS), and the Sounding of the Atmosphere using Broadband Emission Radiometry (SABER) offers very good global

coverage, but relatively low vertical resolution (Maycock et al., 2018). The Global Navigation Satellite System – Radio Occultation (GNSS-RO) data also have very good global coverage and, in addition, offer a very high vertical resolution. GNSS-RO data are available for the period starting in 2002 thus restricting trend estimates to the last 20 years. TTL temperature measurements from radiosonde stations extend back in time much longer; however, climate records of radiosonde temperature often suffer from time-varying biases or inhomogeneities caused by changes in measurement practices (Seidel and Randel,

2006; Wang et al., 2012). Reanalysis models offer TTL temperature records with different vertical resolutions (Tegtmeier et al., 2020a). While meteorological reanalysis data are often utilized as "stand-ins" for observations, trends in reanalysis fields can be impacted by spurious changes in the quality and quantity of the observations used as input data (e.g., Long et al., 2017). Over the past several decades the global mean stratosphere has cooled due to increasing GHGs and decreases in stratospheric ozone (e.g., Maycock et al., 2018). The onset of ozone recovery in the late 1990s has led to a reduction in ozone-induced

cooling with stratospheric cooling trends over 1998–2016 compared to 1979–1997. The troposphere, on the other hand, is warming due to the increasing GHGs. Amplified warming of the upper tropical troposphere compared to the surface (Santer et al., 2005), the so-called tropical amplification is driven by convective process and water vapor feedback as well as oceanic dynamical process and heat storage (e.g., Song et al., 2014; Li et al., 2019). Trends in TTL temperature vary between cooling and warming depending on the time period considered, season, and region. Reanalysis data for 1979-2014 suggest zonally

asymmetric cold point temperature trends with positive trends of 0.22 K/decade over the tropical central and eastern Pacific and negative trends of -0.08 K/decade throughout the remaining tropical regions (Hu et al., 2016). Zonal mean temperature trends from reanalysis over 2002–2020 suggest small but substantial cooling of -0.3 to -0.6 K/decade at 100 hPa and the cold point, which are statistically compatible with trends based on the adjusted radiosonde data sets (Tegtmeier et al., 2020a). Based



on MSU and AMSU satellite records, Garfinkel et al. (2013) showed that there has been a strong warming trend in the tropical
upper troposphere near the Indo-Pacific warm pool, while the trends in the western and central Pacific are much weaker. They
found these patterns to be opposite to the lower stratosphere trends, where the cooling trend is strongest over the Indo-Pacific
warm pool and weaker over western and central Pacific waters. GNSS-RO data available for the last two decades have revealed
a continued strong upper tropospheric warming extending into the lower tropical stratosphere (Ladstädter et al., 2023).

Water vapor measurements in the upper troposphere and lower stratosphere are available from the Microwave Limb Sounder
(MLS) at near-global coverage starting in 2004. The Stratospheric Water and OzOne Satellite Homogenized (SWOOSH)
dataset combines water vapor data from multiple instruments (Davis et al., 2016). Based on these two data sets a significant
moistening of the stratosphere was found starting in 2002 and with relatively high values persisting over the last decade
(Konopka et al., 2022). This moistening is prominent in late boreal winter and spring and shows a clear hemispheric
asymmetry, with the largest positive values in the Northern Hemisphere and negative trends in the polar Southern Hemisphere.
The eruption of the Hunga Tonga–Hunga Ha'apai volcano (Tonga) in January 2022 resulted in a substantial injection of water
vapor directly into the upper atmosphere (Millán et al., 2022; Vömel et al., 2022) which will impact estimates of stratospheric
water vapor trends that go beyond the end of 2021.

Here, we update available long-term trends of cold point tropopause temperatures from GNSS-RO data and of stratospheric
water vapor from MLS and SWOOSH data. We investigate spatial and seasonal patterns of the cold point temperature trends
and their relation to tropospheric and stratospheric changes in Sect. 3. The impact of the seasonality of the cold point
temperature trends on the seasonality of water vapor trends and specifically on the tape recorder signal are presented in Sect.
4. Data sets and methodology are introduced in Sect. 2, and a discussion and summary of the results are provided in Sect. 5.

## 2 Data and Methods

### 2.1 GNSS-RO temperature data

The Radio Occultation (RO) data derived from the Global Navigation Satellite System (GNSS) exhibit widespread coverage,
remain unaffected by clouds and precipitation, and boast exceptional vertical resolution and accuracy, as highlighted by
Kursinski et al. (1997). In this study, we leverage GNSS-RO measurements collected from various satellite missions, including
CHAMP (Wickert et al., 2001), COSMIC (Anthes et al., 2008), GRACE (Beyerle et al., 2005), Metop-A (Von Engeln et al.,
2011), successive Metop-B, METOP-C, SAC-C (Hajj et al., 2004), TerraSAR-X (Beyerle et al., 2011), and KOMPSAT-5
(Bowler, 2018).

The dataset employed in this analysis represents an updated version of the data used in Tegtmeier et al. (2020a). Specifically,
we utilize a dataset encompassing monthly mean CPT temperature, height, and pressure data, along with temperature values
at various pressure levels (30, 50, 60, 70, 100, 150, 200, 300, and 400 hPa). These data are sourced from all GNSS-RO missions
and are spatially gridded on a $30° \times 10°$ grid within the latitudinal range of 30° N to 30° S, covering the period from 2002 until
October 2022.




## 2.2 MLS water vapor data

The Microwave Limb Sounder (MLS) instrument onboard the Aura spacecraft is a part of NASA's Earth Observing System (EOS) platform launched in 2004. MLS uses the microwave limb sounding technique to measure vertical profiles of many atmospheric constituents from the upper troposphere through the mesosphere (Waters et al., 2006). In this study, we use MLS/Aura Level 3 monthly binned water vapor ($H_2O$) mixing ratios, data version 5.1 (Lambert et al., 2020) during 2004-2021. The spatial coverage is near-global (82°S to 82°N) at a resolution of 4° latitude by 5° longitude. A recently identified drift in the MLS v4 $H_2O$ has been corrected in version 5 and as a result, the MLS v5 $H_2O$ record shows no statistically significant drifts compared to ACE-FTS and reduced drifts when compared to balloon-borne frost point hygrometer measurements (Livesey et al., 2021). We omit data after the end of 2021 to avoid having the long-term changes impacted by the anomalous entrainment of water vapor with the Hunga Tonga–Hunga Ha'apai eruption.

## 2.3 SWOOSH water vapor data

The Stratospheric Water and OzOne Satellite Homogenized (SWOOSH) v2.6 data set is a merged record of stratospheric ozone and water vapor measurements taken by a number of limb sounding and solar occultation satellites over the previous ~30 years (Davis et al., 2016). The SWOOSH record spans 1984 to the present and is comprised of data from the SAGE-II/III, UARS HALOE, UARS MLS, and Aura MLS instruments (v4.2). The measurements are homogenized by applying corrections that are calculated from data taken during time periods of instrument overlap. The SWOOSH data product consists of monthly mean zonal-mean values on a pressure grid at a resolution of 5° latitude by 20° longitude.

## 2.4 Regression Method

In order to isolate tropospheric and stratospheric drivers of TTL temperature variability from long-term changes, a multiple linear regression analysis was used. The regression was applied to the CPT, 30, 50, 60, 70, 100, 150, 200, 300, and 400 hPa time series of deseasonalized monthly mean temperature anomalies for each 30°×10° grid cell and tropical mean values. The analysis isolates the linear long-term changes from temperature variability driven by QBO, ENSO, and SAOD based on the regression described in Eq. (1).

$$T(t) = A + B \cdot t + C_1 \cdot QBO_1(t) + C_2 \cdot QBO_2(t) + D \cdot ENSO(t) + E \cdot SAOD(t) + \epsilon(t) \,, \tag{1}$$

where $T(t)$ are the temperature anomalies and $\epsilon(t)$ are the residuals. The output of the regression provides the linear trend in the form of the regression coefficient $B$. The $QBO_1(t)$ and $QBO_2(t)$ terms are orthogonal time series representing QBO variations constructed as the first two empirical orthogonal functions of the Freie Universität Berlin (FUB) radiosonde stratospheric winds (Naujokat, 1986; Wallace et al., 1993). $ENSO(t)$ is the multivariate ENSO index (https://www.esrl.noaa.gov/psd/enso/mei/), and $SAOD(t)$ is the detrended stratospheric aerosol optical depth from the Global Space-based Stratospheric Aerosol Climatology (Thomason et al., 2018). Significance is tested based on a two-tailed test with a 95% confidence interval.



# 3 Cold point temperature trends

The long-term change in tropical mean CPT temperature has been calculated for GNSS-RO 2002-2022 data (Fig. 1) by applying the multiple linear regression described in section 2.4. The deseasonalized temperature anomalies (blue line), used

as input data to the regression analysis, exhibit strong interannual variations with a range of approximately ±2K. The R-squared value of 0.3 indicates that approximately 30% of the variability in CPT temperature variations can be explained by the long-term trend, QBO, ENSO, and SAOD. The QBO results in periodic oscillations in CPT temperature, albeit with a relatively modest magnitude (reddish line). ENSO variations in TTL temperature are slightly larger (purple line), showing often, but not always, warm anomalies coinciding with positive ENSO phases (e.g., 2011 and 2021-2022). The fluctuations in SAOD are

relatively minor. After taking into account these factors, the regression suggests a positive linear trend (black line) of 0.36 ± 0.06 K/decade. It should be emphasized that substantial residuals (not shown here) indicate a significant portion of variability that cannot be accounted for by the prescribed regressors. This variability may arise from non-linear processes or processes not captured by the proxies, such as fluctuations in tropical upwelling with the BDC due to extratropical wave driving and changes in convective activity not associated with ENSO.


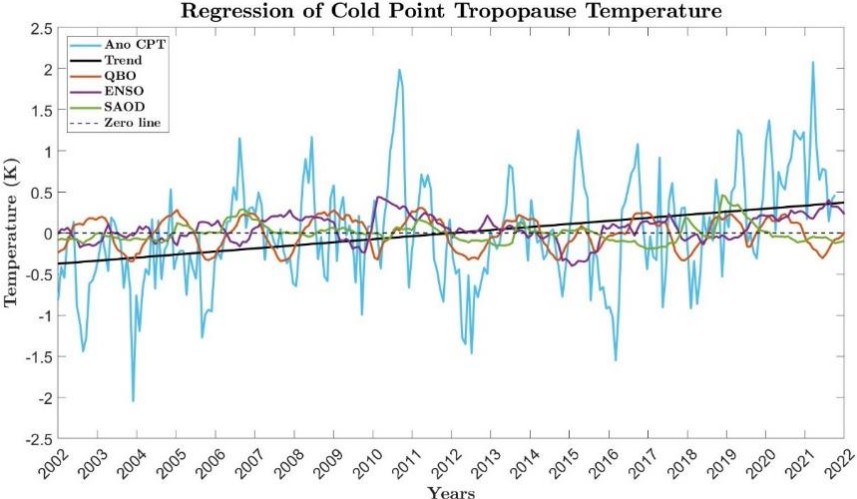

**Figure 1. Tropical mean (20°S-20°N) deseseasonalized cold point tropopause (CPT) temperature anomalies (Ano CPT), regression terms (QBO, ENSO, SAOD), and linear trend based on GNSS-RO data.**

In the second step, spatial and seasonal patterns of CPT temperature trends have been analyzed based on the regression analysis

applied to seasonal and monthly temperature time series at each grid point. The findings, depicted in Fig. 2, indicate that over the past two decades, there has been no overall cooling trend of the CPT, in contrast to observations prior to 2000 (e.g., Randel et al., 2006; Rosenlof and Reid, 2008). For the full monthly mean time series, clear warming of the CPT with the strongest and significant trends over the tropical regions of South America, the Atlantic, Africa, the Indian Ocean, and Australia of up to 0.71 K/decade can be observed (Fig. 2a). Temperature trends minimize to around zero over the central Pacific region.





Seasonal mean CPT temperature trends (Fig. 2b-e) are similar to trends based on the full-time series. Temperature changes in the Northern Hemisphere (NH) winter (December, January, and February, DJF), spring (March, April, and May, MAM), summer (June, July, and August, JJA), and fall (September, October, and November, SON) indicate prevalent CPT warming across the tropics. The warming is especially pronounced over the tropical Atlantic and South America in NH winter and spring (up to 1.13 K/decade) as well as Central Africa and the Maritime Continent in NH spring (up to 0.91 K/decade). During

NH summer and fall, the cold point tropopause shows weak and non-significant warming trends except for Australia, which shows significant cold point warming (up to 1.25 K/decade). Similar results are found for NH fall with mostly non-significant cold point trends except for warming over the Atlantic (up to 0.94 K/decade).

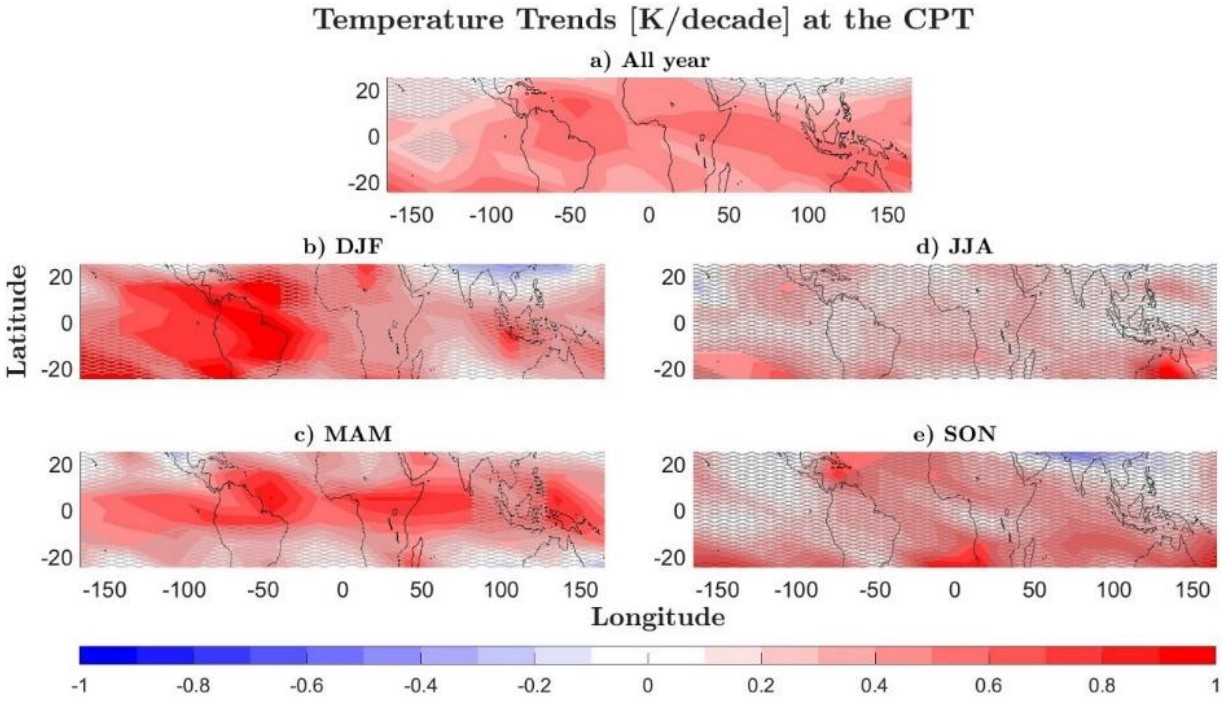

**Figure 2. Long-term (2002-2022) temperature trends [K/decade] of the cold point tropopause (CPT) temperature based on the GNSS-RO dataset for the monthly mean (a) and seasonal mean time series (b-e). Trends not significant at the 95% confidence level are marked with grey crosses.**

Trends of cold point height and pressure can be expected to be linked to long-term changes in CPT temperature. A comparison of the seasonal trends in CPT temperature (Fig. 2b-e), height (Fig. 3a-d), and pressure (Fig. 3e-h) reveals a noteworthy

consistency of the seasonality of the signals. Seasons with the strongest warming (DJF and MAM) show a decrease in CPT height (up to -0.12 km/decade) and an increase in CPT pressure (up to 2.4 hPa/decade). Regional patterns between temperature and height trends are similar for MAM, but less so for DJF. However, these trends in cold point pressure and specifically in





cold point height are significant only over a few smaller regions. During NH summer and fall, most regions experience a rise of the CPT (positive trend in height and negative trend in pressure) while temperature showing weak warming trends.

The investigation of the seasonal cycle and the vertical structure of TTL temperature trends highlights discernible patterns (Fig. 4). Overall, the well-known upper tropospheric warming extends into the lower stratosphere up to 80 hPa during the NH spring and summer, 50 hPa during NH winter and all the way up to 30 hPa during October. Temperature trends at the cold point show a clear seasonal cycle with a pronounced warming during boreal winter and spring and weaker, non-significant warming for the rest of the year. For the upper troposphere, the seasonal cycle of the temperature trends is nearly opposite to

that of the CP, with largest warming during NH late summer and winter.

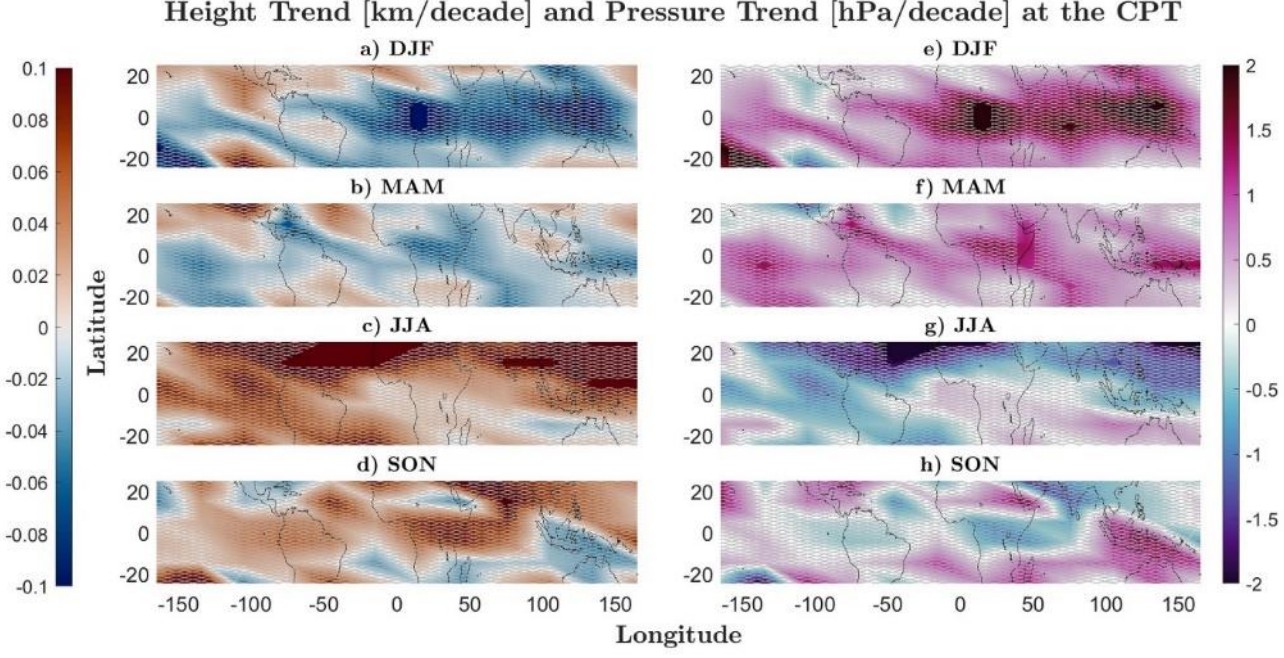

**Figure 3. Cold point tropopause (CPT) height [km/decade] (left column) and pressure [hPa/decade] (right column) trends for each season during 2002-2022 based on GNSS-RO data. Trends not significant at the 95% confidence level are marked with gray**
**crosses.**





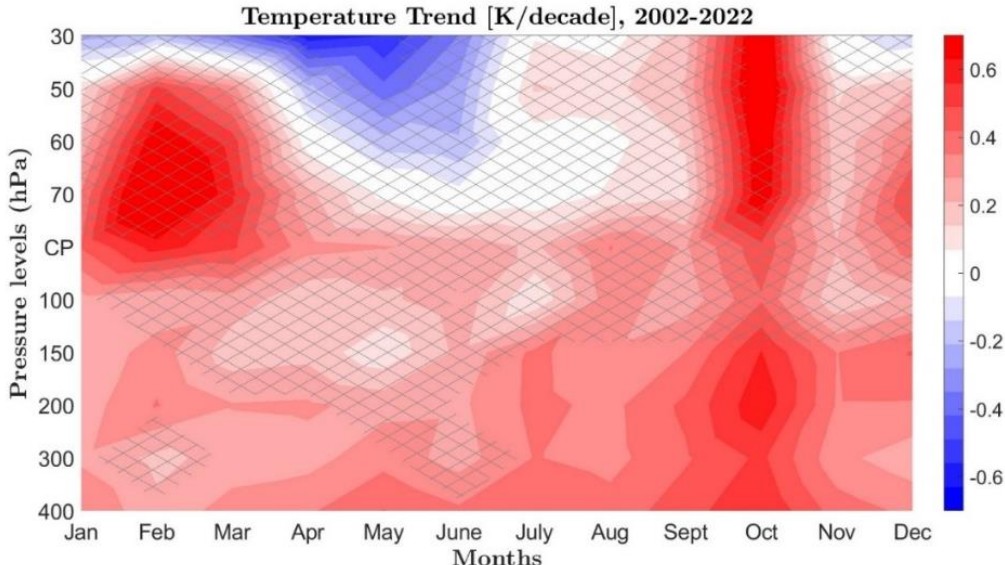

**Figure 4. The seasonal cycle of temperature trends [K/decade] for 2002-2022 at the cold point (CP) and different pressure levels based on GNSS-RO data. Trends not significant at the 95% confidence level are marked with crosses.**

The connections between temperature trends at the cold point trends and in the lower stratosphere and upper troposphere (at fixed pressure levels at 30, 50, 70,100, 200, 300, and 400 hPa) are analysed by comparing their spatial patterns (Fig. 5). Focusing on boreal winter trends at various pressure levels, the findings highlight a notable pattern: in regions where the cold point exhibits the strongest warming (specifically, the tropical Atlantic and South America), temperatures in the upper troposphere (400 and 300 hPa) experience a slight cooling trend, although this lower-level cooling trend is not statistically

significant. Conversely, regions characterized by weaker and nonsignificant cold point cooling such as parts of East Asia demonstrate substantial warming in the upper troposphere. Such reversed dipole patterns are also observed during the boreal fall season when the cold point shows some local cooling over the tropical East Pacific and East Asia (Fig 5, approximately 90–140°E and 10°N). This cooling is also evident at 70 and 100 hPa, while in the upper troposphere at 300 and 400 hPa the same regions show the strongest warming trends. During MAM and JJA, the observed trends do not exhibit significant dipole

patterns or clear trend reversals between the CP and the upper troposphere. Instead, these seasons predominantly display warming trends across most tropical regions and all levels between CP and 400 hPa (not shown here).





**Figure 5. Temperature trends [K/decade] during DJF and SON 2002-2022 at different pressure levels and the cold point (CP) based on GNSS-RO dataset. Trends not significant at the 95% confidence level are marked with grey crosses.**





The connections between temperature trends at the cold point and in the upper troposphere as well as lower stratosphere are further analyzed by calculating the Pearson correlation coefficients between the gridded trends at the CP and the trends at all other levels, respectively (Fig. 6). Notably, the cold point temperature trend patterns exhibit striking similarities with the temperature trend patterns at levels slightly below (100 hPa) and above (70 hPa) the CPT. These similarities extend upwards into the lower stratosphere (60 to 50 hPa) during NH winter. The connection between temperature trends at the cold point and

in the upper troposphere manifests itself in the form of clear anti-correlation between the trend patterns, particularly during DJF and SON seasons as discussed above. Anticorrelations maximize at 300 hPa for DJF and 400 hPa for SON. This suggests that the intensity of cold point warming is connected to patterns of enhanced and reduced upper troposphere warming via some regional dynamical or radiative processes.

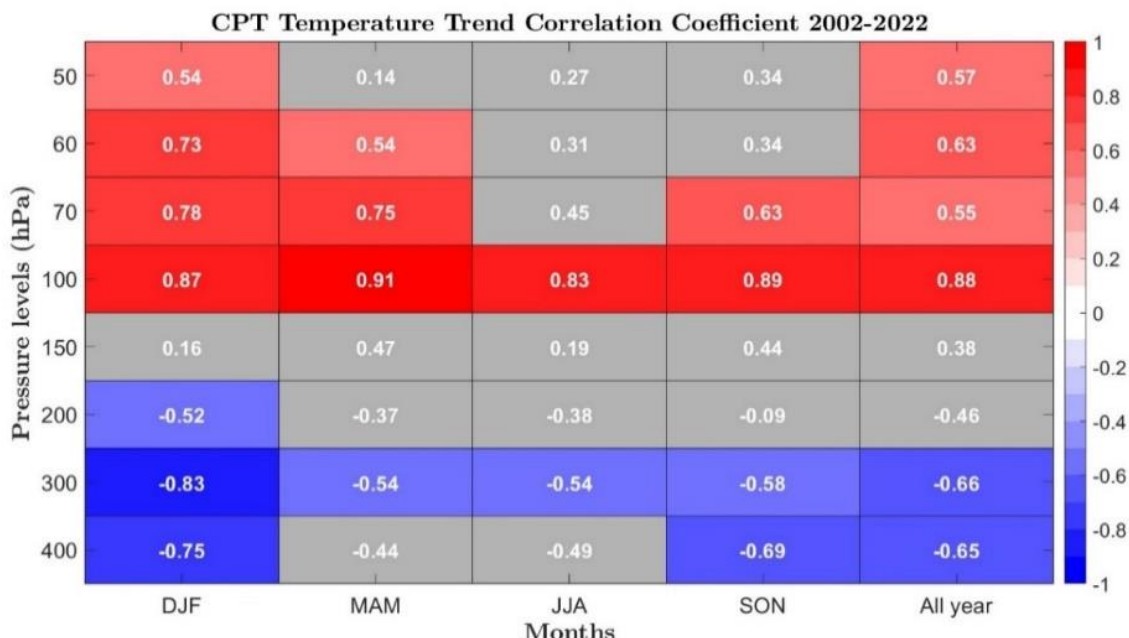

**Figure 6. Correlation coefficients between gridded temperature trends at the CPT and at different pressure levels as given on the y-axis based on GNSS-RO data for 2002-2022. Correlation coefficients in grey boxes are not significant at the 95% confidence level.**

## 4 Stratospheric water vapor trends

    The temperature of the cold point plays a crucial role in governing the water vapor content of the lower stratosphere. In order to examine the influence of the spatial variations in cold point temperature trends, the patterns of CPT temperature and water

vapor trends are compared in Fig. 7 for the 2004-2021 time period. The analyzed data sets, SWOOSH and MLS, agree on the overall trend patterns, with some differences for individual regions and with respect to the trend significance. These differences are largely driven by the fact that SWOOSH uses MLS v4.2, which is compared here to MLS v5. While the stratosphere shows



overall moistening, these positive trends are only significant over South America for DJF and MAM and over the Maritime

Continent for MAM for SWOOSH. Spatial variations in temperature trends at the CPT (and 70 hPa, not shown here) are found

to coincide with spatial variations of water vapor trends at 100 hPa (and 82 hPa, respectively, not shown here). The strongest

moistening is observed over the regions that show the largest warming trends (e.g., South America in DJF). Regions with small

cold point cooling, on the other hand, show weak and not significant drying (e.g., Southeast Asia in DJF).

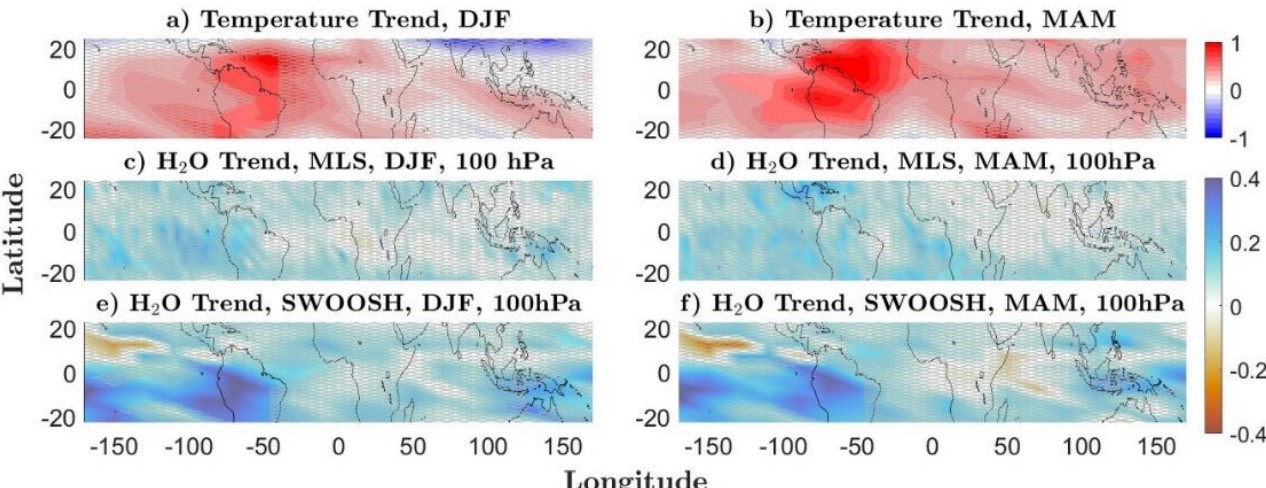

**Figure 7. Temperature trends [K/decade] at the cold point tropopause based on GNSS-RO data (a-b), and water vapor trends**
**[ppm/decade] by MSL data and SWOOSH data at 100 hPa (c-f) for DJF and MAM 2004-2021. Trends not significant at the 95%**
**confidence level are marked with grey crosses.**

Given the strong seasonal cycle in cold point temperature trends, one can expect water vapor trends to have similar seasonal

variations. Fig. 8 shows the seasonal cycles of cold point temperature and 100 hPa water vapor trends for 10°S-10°N. The cold

point displays the strongest warming during the boreal winter and spring months with significant trends from February to June

(see also Fig. 4). Most of the boreal summer and fall months, on the other hand, experience weak and non-significant warming.

This seasonality of the CPT temperature trend extracted from GNSS-RO aligns closely with the seasonality found for the water

vapor trends sourced from MLS and SWOOSH data. Consistent with peak warming in boreal winter and spring, water vapor

at 100 hPa shows the strongest positive trends during the same time of the year and small, non-significant trends during boreal

autumn. It is worth noting that both water vapor datasets indicate a maximum trend slightly later than the maximum cold point

warming. The reasons for this discrepancy are currently unclear and may be related to limitations of the vertical resolution of

the measurements or involve other processes potentially influencing water vapor trends during boreal spring.

MLS water vapor trends are smaller than SWOOSH trends but agree with their uncertainties. MLS water vapor trends also

show weaker seasonal fluctuations (amplitude of 0.1 ppm/decade) compared to SWOOSH (0.2 ppm/decade). The seasonal



cycle of the cold point warming shows an amplitude of 0.36 K/decade. The ratios of the water vapor and temperature trend
amplitudes of 0.3 ppm/K and 0.6 ppm/K for MLS and SWOOSH, respectively, are consistent with the Clausius-Clapeyron
equation as first shown in Konopka et al. (2022).

**Figure 8.** Seasonal cycle of cold point tropopause (CPT) temperature climatology (brown line) and trend (blue line) based on GNSS-
RO data (a), and water vapor climatology (brown line) and trends (blue line) at 100 hPa (b-c), and 21 hPa (d-e) based on MLS and
SWOOSH data for 2004-2021. Error bars show ±2-sigma confidence intervals of the trend.

The stronger moistening during boreal winter and spring when TTL water vapor shows minimum values (Fig. 8b-c), along
with the weaker moistening during NH fall when water vapor shows maximum values, suggest a weakening of the water vapor
seasonal cycle. The same principle applies to the seasonal cycle in cold point temperatures where the strongest warming





coincides with the season of lowest temperatures (Fig. 8a) thus leading to a decrease of the seasonal cycle amplitude over time.

As the cold point and water vapor seasonal cycles set the entry values of the tape recorder (Mote et al., 1996), their weakening can lead to a weakening of the tape recorder signal. Indeed, the seasonal cycle of water vapor trends in the middle stratosphere (38 hPa, Fig. 8 d-e) illustrates how the signal propagates upward in time with the strongest moistening now appearing during boreal late fall and winter. As the climatological seasonal cycle propagates upward in the same way, the largest moistening still coincides with the timing of the lowest water vapor values while maximum water vapor experiences weaker moistening.

The upward transport of the deseasonalized water vapor anomalies is shown in Fig. 9a and c for MLS and SWOOSH data over 10°N-10°S. All time-series are boxcar smoothed (width = 5 months) to remove sub-seasonal variability. The mean seasonal cycle of the tape recorder shows the expected minimum during NH winter and spring followed by an increase during summer and fall at 100 hPa and the slow upward propagation of this pattern to higher levels. Water vapor trends have been calculated based on the multilinear regression for each level and month of the year for MLS and SWOOSH (Fig. 9b and d). Noticeably,

the findings illustrate an upward transport of the trends in water vapor at similar time scales as the upward transport of the seasonal cycle in water vapor. The black dotted lines illustrate the timing of the upward transport of water vapor seasonal maxima and minima. The same black lines are overlaid on top of the water vapor trend demonstrating that the minimum (maximum) in water vapor coincides with the stronger (weaker) moistening on most levels.

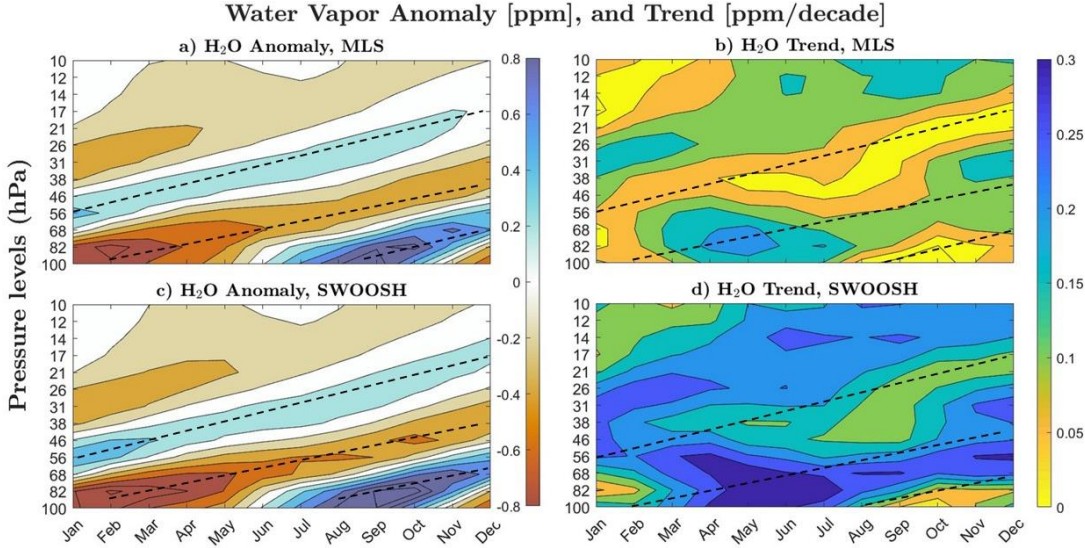

**Figure 9. Deviation in water vapor [ppm] from the time mean averaged profile averaged over 10°N-10°S for MLS (a), and SWOOSH (c) data for 2004-2021. Upward transport of seasonal maxima and minima are shown as black dotted lines. Water vapor trend [ppm/decade] from MLS (b), and SWOOSH (d) data with black dotted lines from panel a and c overlaid on top.**

The percentage change in amplitude of the water vapor seasonal cycle based on the long-term linear trend has been calculated by assuming that the climatological seasonal cycle changes only according to the linear trend for the respective time period

(Table 1). At 100 hPa, the tape recorder signal is estimated to decrease by as 5 to 6% per decade. Such decreasing amplitude





is also found at most other levels consistent with our hypothesis of upward propagation of the weakening. An exception is the levels 56 hPa and 46 hPa which show a slightly increasing amplitude, inconsistent with trends at the other levels. The weakening of amplitude relative to the background seasonal cycle changes with increasing altitude, in particular above 30 hPa, possibly due to the influence of other seasonal signals in the middle stratosphere.

**Table 1. Change in amplitude of the tropical water vapor tape recorder [% per decade] averaged over 10°N-10°S for MLS and SWOOSH data for 2004-2021.**

| Pressure levels (hPa) | SWOOSH data | MLS data |
| --- | --- | --- |
| **21** | -23.8 | -24.4 |
| **26** | -27.8 | -25.6 |
| **31** | -15.5 | -15.9 |
| **38** | -13.6 | -16 |
| **46** | 1.9 | -4.2 |
| **56** | 4.8 | 1.8 |
| **68** | -2.7 | -6.2 |
| **82** | -0.05 | -4.4 |
| **100** | -5.7 | -5.2 |

## 5 Discussion and summary

High vertical resolution GNSS-RO data demonstrates that over the past two decades, there has been no overall cooling trend in the TTL region consistent with previous studies (e.g., Randel et al., 2017; Steiner et al., 2020; Ladstädter et al., 2023).
Focusing on the cold point, which sets the entry values of stratospheric water vapor, our study shows clear warming of this level with strongest trends over the tropical regions of South America, the Atlantic, Africa, the Indian Ocean, and Australia of up to 0.7 K/decade. This cold point warming over 2002-2022 is in contrast to observations prior to 2000, which show a clear cold point cooling (Randel et al., 2006; Wang et al., 2012). In addition, we find an upward shift of the cold point during boreal summer and autumn and a downward shift during boreal winter and spring in good agreement with results derived from the
ozonesonde network SHADOZ (Southern Hemisphere Additional Ozonesondes; Thompson et al., 2021).
Modelling studies have suggested warmer and higher tropopause levels under future global warming due to local radiative effects from increased GHGs and due to a warmer troposphere (Kim et al., 2013; Lin et al., 2017). This temperature signal agrees with the here identified cold point warming, suggesting that radiative effects drive the trends observed over the last 20 years. However, our tropopause height trends with a downward shift during the seasons of maximum warming (DJF and
MAM) are not consistent with the model results. This implies that dynamical processes might also play a role for the observed cold point tropopause trends as discussed below.





Upper tropospheric warming is known to be an amplification of the surface warming signal maximizing in the tropics around 200 hPa and is predicted to decrease to zero near the tropical tropopause (e.g., Santer et al., 2005; Keil et al., 2023). The strong upper tropospheric warming is evident from the GNSS-RO observations (Fig. 4 and 5) and agrees well with estimates from

other observational data sets (Vergados et al., 2021; Steiner et al., 2020). Previous studies have suggested that observed temperature trends in the tropical upper troposphere and in the lower stratosphere are anti-correlated (Garfinkel et al., 2013; Hu et al., 2016) in agreement with our results. While 1979–2014 reanalysis datasets suggest strongest warming over the tropical central and eastern Pacific (Hu et al., 2016), our analysis of 2002-2022 GNSS-RO data finds the warming signal to minimize over the central Pacific and to maximize over the Atlantic (Fig. 2). The regions of strongest tropopause warming reveal slight

cooling trends in the upper troposphere while regions characterized by weaker and nonsignificant tropopause cooling demonstrate substantial warming in the upper troposphere (Figs. 5 and 6). The identified anticorrelations suggest that dynamical responses to the enhanced moist heating in the upper troposphere or SST driven enhanced convection can be expected to result in negative cold point trends potentially cancelling some of the radiatively driven warming.

Global stratospheric cooling has weakened over the last two decades due to the reduction in ozone-induced cooling after the

onset of ozone recovery (Randel et al., 2017; Maycock et al., 2018). However, the tropical lower stratosphere continues to experience negative ozone trends (e.g., Godin-Beekmann et al., 2022), which leads to ongoing cooling and is therefore not consistent with the warming of the tropical lower stratosphere extending up to 30-70 hPa (Fig. 4). Dynamical changes related to upwelling with the stratospheric Brewer-Dobson circulation (BDC), or local impacts of wave activity can also impact tropopause trends. Tropical TTL warming could result from decreased upwelling due to a weakening of the BDC. While this

weakening is not expected from model projections under increasing GHGs, some observational data sets suggest positive, but non-significant age-of-air trends in the TTL for 2002-2012 (Haenel et al., 2015) indicating weakened BDC transport in this region. Global lower-stratospheric temperature observations from MSU/AMSU also suggest that the annual mean BDC accelerated from 1980–1999 and decelerated from 2000–2018 (Fu et al., 2019) consistent with changes in TTL temperatures from GNSS-RO data identified here. We find the strongest cold point warming during boreal winter which would be consistent

with a more pronounced weakening of the BDC during this time of year. In addition to boreal winter, the cold point and lower stratosphere show a strong warming signal in October, which has also been suggested to be related to the weakening of the BDC during this month as it coincides with a strong negative temperature trend at the southern hemisphere polar regions (Ladstädter et al., 2023).

The cold point temperatures have a clear seasonal cycle with the lowest temperatures during boreal winter and spring due to

increased upwelling with BDC. The cold point temperature trends also show a seasonal cycle with the strongest warming during boreal winter and spring. Weaker warming trends found for the rest of the year were not significant due to the strong variability of the temperature time series. We find that the amplitude of the temperature seasonal cycle decreased by 7% as a result of the seasonal alignment of the coldest temperatures with the strongest warming.

Water vapor observations at 100 hPa, a level very close to the cold point, show positive trends for 2004-2021 throughout most

of the tropics consistent with increasing cold point temperatures. This is consistent with multi-decadal variation in water vapor



identified for a longer time series extending over the past 40 years (Tao et al., 2023). The positive trend occurred after a sharp drop in water vapor in 2001 (Konopka et al., 2022). The authors showed that the tropical moistening after 2000 occurred mainly during boreal winter and spring, consistent with our results and the seasonal cycle of the cold point warming (Fig. 8). We find that the seasonal cycle in water vapor at 100 hPa is reduced by 5-6% (Table 1) as a result of strongest moistening and

maximum warming during the season of lowest water vapor and coldest temperatures. Our analysis shows that this reduction in the seasonal cycle is transported upwards together with the seasonal anomalies and has reduced the amplitude of the well-known tape recorder over the last two decades.

Overall, the here identified warming of the cold point is consistent with model simulations which attribute the warming trends to radiative effects. Seasonal signals and zonal asymmetries in cold point temperature and height trends, on the other hand,

seemed to be related to dynamical responses to the enhanced moist heating in the upper troposphere, changing convection or trends of the stratospheric BDC. These seasonal variations have important implications for cross-tropopause transport of water vapor. We find a weakened seasonal cycle in cold point temperature that leads to a weakened seasonal cycle in stratospheric moistening and thus a reduction of the tape recorder signal.

*Data Availability:* The GNSS-RO data are available from https://cdaac-www.cosmic.ucar.edu/cdaac/products.html. MLS data can be accessed at https://disc.gsfc.nasa.gov/datasets/ML3MBH2O_005/summary, and SWOOSH data can be obtained from https://csl.noaa.gov/groups/csl8/swoosh/.

*Author contribution:* MZ performed the analysis and wrote the manuscript. ST developed the research question and guided the

research process. RPK provided the GNSS-RO data. SMD provided the SWOOSH data. All authors took part in the process of the manuscript preparation.

*Competing interests*: The contact author has declared that none of the authors has any competing interests.

*Financial support*: This research has been supported by the Natural Sciences and Engineering Research Council of Canada (NSERC, grant no. RGPIN-2020-06292).

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
