# Peer review of "Variability and long-term changes of tropical cold point temperature and water vapor"

_EGUsphere, 2024_

## Author Comment (AC1)

Dear referees,

*We thank the reviewers for their valuable comments which have helped us to improve the paper in revision. We have changed the manuscript according to the comments listed below. Most importantly, we have updated SWOOSH v2.6 to v2.7, adjusted all figures and discussions, improved the hatching to display significance, and improved the overall discussion and conclusion section. Comments are reproduced below, followed by our responses in italics.*

**Anonymous Referee #1:**

General:

This paper is well-written and presents new and interesting results. It relies solely on experimental data to establish positive trends in cold point temperatures for the period 2002-2020, subsequently leading to corresponding positive trends in stratospheric water vapor. The paper explores the seasonality of these trends and their zonal patterns, offering a comprehensive analysis. The abstract effectively summarizes these crucial findings. One particularly noteworthy discovery is the observed decrease in tape-recorder amplitude over the last two decades. The findings are supported by clear and well-constructed figures. However, a minor improvement could be made regarding the crosses in the figures that mark statistically nonsignificant trends, as it can be challenging to see them.

> *Thanks for pointing out the quality of the crossing in the figures. We have chosen a different crossing (darker color and reduced the density of lines) and replaced all figures. Regions with significant trends can now be detected more easily.*

Congratulations on this convincing piece of work.
> *Thanks!*

Here, few detailed comments:
Abstract:

L20: ...on the other hand, seems to be related...
> *We have modified the text accordingly.*

L36: As your introduction is very comprehensive, I would also suggest referencing Riese et al, 2012, doi:10.1029/2012JD017751. See their Figure 1 for insight into the motivation behind the extreme radiative sensitivity of the TTL in relation to its composition.
> *We have modified the text accordingly.*

L97: Concerning your sentence: "Zonal mean temperature trends from reanalysis over 2002–2020 suggest small but substantial cooling of -0.3 to -0.6 K/decade at 100 hPa and AT the cold point, which are statistically compatible with trends based on the adjusted radiosonde data sets (Tegtmeier et al., 2020a)." I believe Figure 11 from the cited paper displays results for the 1979-2005 period, not the 2002–2020 period as mentioned in the sentence. Additionally, you later

demonstrate that the trends are predominantly positive, not negative, for the 2002–2020 period. Please provide clarification.

> *This is correct. Tegtmeier et al. show temperature trends for 1979-2005, sorry for the confusion. We have modified the text accordingly.*

L125-135: It would be nice to have 1-2 sentences, how does radio occultation work and how atmospheric temperature can be derived from such observations.

> *We have added the following text to section 2.1 of the manuscript: 'GNSS-RO measurements are obtained when a low earth orbiter (LEO) satellite measures the Doppler shift of the radio signal emitted by GNSS satellites after it travelled through the Earth's atmosphere, which slightly bends the signal's path due to its refractivity gradients. The relative motion between LEO and GNSS satellite orbits enables a vertical scan of the atmosphere. In the upper troposphere and stratosphere, the high-resolution refractivity profile is effectively a function of pressure and temperature, while humidity is increasingly important lower down.'*

L145: I thought, the newest SWOOSH version uses MLS v5.1?

> *Thanks for pointing this out. We realized that we originally used SWOOSH v2.6 (using MLS v4.2) and changed all figures and corresponding text to now be based on SWOOSH v2.7 (using MLS v5.1).*

L153: It would be nice to have the explanation of the abbreviation "SAOD" in this sentence and not first in L159.

> *We have modified the text accordingly.*

L171: Please reformulate: It should be emphasized that substantial residuals (not shown here) indicate a significant portion of variability of around 70%.... You mentioned that only 30% of the variability can be explained by your MLR model. Sometimes, people utilize lagged MLR models, considering that ENSO and SAOD signals take time to propagate from their source regions (ENSO - from the Earth's surface, SAOD -from the maximum of aerosol absorption around 50 hPa) to the region of interest. This might slightly increase the amount of variability that can be explained by the MLR.

> *We have modified the text accordingly.*
> *We have also conducted sensitivity studies using a lagged regression for the ENSO and SAOD terms but found that the overall regression did not improve in a consistent way over all regions and seasons. We have therefore decided to use the regressions without time lag.*

Fig 4: I think, you show mean tropical values (20S-20N). You should mention it both in the caption and in the text.

> *We have modified the text accordingly.*

---

## Author Comment (AC2)

Dear referees,

*We thank the reviewers for their valuable comments which have helped us to improve the paper in revision. We have changed the manuscript according to the comments listed below. Most importantly, we have updated SWOOSH v2.6 to v2.7, adjusted all figures and discussions, improved the hatching to display significance, and improved the overall discussion and conclusion section. Comments are reproduced below, followed by our responses in italics.*

**Anonymous Referee #2:**

This work uses GNSS-RO satellite data, with a very high vertical resolution to study the trend of the tropical tropopause layer (TTL) during the past 20 years and uses SWOOSH homogenized satellite data and MLS satellite data to study the trend of water vapor. This paper is well structured and almost all points are clearly explained. The trend was studied in terms of different location and seasons, and the authors tried to explore the role of radiation and dynamics in the observed trend. I recommend a minor revision before accepting this paper.

Major comment:

The major comment of this paper is regarding the conclusion 'overall observed warming of the cold point is due to radiative effects, and seasonal signals and zonal asymmetries are due to dynamics' in line 383, section 5. I agree with the analysis in section 3, and the conclusion that 'cold point warming is connected to patterns of enhanced and reduced upper troposphere warming via some regional dynamical or radiative processes'. However, this does not necessarily lead to the conclusion in line 383. It could only be drawn when the authors analyzed (1) the seasonality and location of the BDC, (2) the overall trend of the BDC, (3) the seasonality and location of the convection, and (4), the overall trend of the convection. The authors do not provide the location/seasonality/overall trend of the Brewer-Dobson circulation (dynamics), so it is hard to conclude that the BDC are not accounting for the overall trend and instead just the seasonality and location. The authors also listed previous works that concerning the overall BDC trend in lines 354-369, and this may contribute to the overall trend of the TTL temperature. I don't understand why the conclusion is 'it explains the seasonality', instead of the overall trend.

> *Thanks for pointing this out. Indeed, our conclusion here was not correct and we have changed the text so that it now states that the trend is consistent with radiative effects found in models and with a weakening of the BDC derived from satellite observations. The last paragraph of the discussion and summary now states:*
>
> *'Overall, the here identified warming of the cold point is consistent with model simulations which attribute the warming trends to radiative effects. At the same time, the cold point warming is consistent with a potential weakening of the BDC upwelling in the tropical lower stratosphere. Some observations such as satellite temperatures suggest such weakened upwelling; however, these trends show large uncertainties and disagree with model simulations. Seasonal signals and zonal asymmetries in cold point temperature and height*

*trends might be related to dynamical responses to the enhanced moist heating in the upper troposphere, changing convection or variations in the trends of the stratospheric BDC.'*

Minor comments:

Line 76: the Brewer-Dobson circulation also plays an important role, see major comments.
  *We have modified the text accordingly.*

Line 81: "A very high vertical resolution": it will be better if the authors provide an estimated number of the vertical resolution.
  *We have added the following sentence to section 2.1: 'The vertical resolution of GNSS-RO temperature data is variable, ranging from ∼1 km in regions of constant stratification down to 100–200 m where the biggest stratification gradients occur, e.g., at a very sharp tropopause (Gorbunov et al., 2004)'.*

Line 129: 30°x10°: please clarify what is latitude and what is longitude.
  *This is a 30°×10° longitude-latitude grid. We have modified the text accordingly.*

Line 153: SAOD: this is the first time that the term "SAOD" is used, please define this abbreviation.
  *We have modified the text accordingly.*

Line 252: SWOOSH result is very different from MLS, which uses the MLS 4.2 with a drift problem, but it is also a homogenized data and has its strength when comparing with MLS. How much of the difference between SWOOSH is due to MLS 4.2, and how much is from other satellite data? For example, the authors can compare the trend of SWOOSH data, MLS 4.2, and MLS 5.0.

  *Thanks for pointing this out. We realized that made a mistake by including SWOOSH v2.6 (which uses MLS v4.2). Given that MLS4.2 has the known drift problem, we prefer to include SWOOSH v2.7 (which uses MLS v5.1) and changed all figures and corresponding text accordingly. Differences that can be seen in the new version of our figures can be attributed to SWOOSH being a homogenized data set that also uses information from other satellite instruments and is given at a different horizontal resolution. Trends in SWOOSH and MLS show similar values and spatial/seasonal patterns with SWOOSH trends being smoother with smaller peak values.*

Line 309: "due to the influence of other seasonal signals in the middle atmosphere": like CH4?

  *Indeed, this could be expected to be one of the most important factors. We have added 'such as the variability of water vapor production from methane' to the sentence.*

Line 385: 'seemed to be' is too vague. See major comments.

  *We have changed the last paragraph of the discussion section including this sentence, see answer above.*

---

## Author Response (AR2)

*Dear Editor,*

*We express our gratitude to the editor for their invaluable comments, Comments are reproduced below, followed by our responses in italics.*

L85: Reanalysis models -> reanalyses

*We have modified the text accordingly.*

L92 and 93: process -> processes

*We have modified the text accordingly.*

L94: Which reanalysis are you refering to here? Related on L97: how many reanlyses contribute to the stated range?

*The statement in line L94 related to Hu et al. (2016) refers to two reanalyses (MERRA and ERA-Interim).*

*The statement in line L97 related to Tegtmeier et al. (2020a) refers to five reanalyses (JRA-25, JRA-55, MERRA, MERRA-2, and CFSR.*

*We have added the number of data sets on both cases.*

L160: Student's t-test?

*Yes, Student's t-test. We have added the information.*

L242: Are you able to speculate on the drivers here? Any indications from literature?

*We have added the following text to the document.*

*The here found anticorrelations of the trend patterns resemble previously observed links between tropospheric and lower-stratospheric temperature anomalies, which have several possible theoretical explanations. On the one hand, cooling of the tropopause has been suggested to result either from convective overshooting of the level of neutral buoyancy (Kuang and Bretherton 2004) or the formation of convective cold tops via hydrostatic adjustment above tropospheric convective heating (Holloway and Neelin, 2007). Alternatively, theoretical evidence has suggested that tropospheric thermal forcing and thus generated tropopause geopotential anomalies can modulate stratospheric upwelling and temperature (Lin and Emanuel, 2024).*

L271: How would vertical resolution limitations cause shifted seasonality of trends?

*We find that the seasonal cycle of the water vapor trends shifts with altitude (see Figure 9b and d) and might therefore be impacted by the vertical resolution of the measurements and satellite instrument averaging kernels.*